# Determinants of HIV testing among Filipino women: Results from the 2013 Philippine National Demographic and Health Survey

Veincent Christian F. Pepito[1,2]*, Sam Newton[1,3]

1 Faculty of Epidemiology and Population Health, London School of Hygiene and Tropical Medicine, University of London, London, United Kingdom, 2 Center for Research and Innovation, School of Medicine and Public Health, Ateneo de Manila University, Pasig City, Philippines, 3 School of Public Health, Kwame Nkrumah University of Science and Technology, Kumasi, Ghana

* vcfpepito12345@gmail.com

## Abstract

### Background

The prevalence of having ever tested for HIV in the Philippines is very low and is far from the 90% target of the Philippine Department of Health (DOH) and UNAIDS, thus the need to identify the factors associated with ever testing for HIV among Filipino women.

### Methods

We analysed the 2013 Philippine National Demographic and Health Survey (NDHS). The NDHS is a nationally representative survey which utilized a two-stage stratified design to sample Filipino women aged 15–49. We considered the following exposures in our study: socio-demographic characteristics of respondent and her partner (i.e., age of respondent, age of partner, wealth index, etc.), sexual practices and contraception (i.e., age at first inter-course, condom use, etc.), media access, tobacco use, HIV knowledge, tolerance to domestic violence, and women's empowerment. The outcome variable is HIV testing. We used logistic regression for survey data to study the said associations.

### Results

Out of 16,155 respondents, only 372 (2.4%) have ever tested for HIV. After adjusting for confounders, having tertiary education (adjusted odds ratio (aOR) = 2.15; 95% Confidence Interval (CI): 1.15–4.04), living with partner (aOR = 1.72; 95% CI: 1.19–2.48), tobacco use (aOR = 1.87; 95% CI: 1.13–3.11); belonging to the middle class (aOR = 2.72; 95% CI: 1.30–5.67), richer (aOR = 3.00; 95% CI: 1.37–5.68), and richest (aOR = 4.14; 95% CI: 1.80–5.91) populations, having weekly television access (aOR = 1.75; 95% CI: 1.04–2.94) or internet access (aOR = 2.01; 95% CI: 1.35–3.00), living in a rural area (aOR = 1.87; 95% CI: 1.34–2.61); and being a Muslim (aOR = 2.30; 95% CI: 1.15–4.57) were associated with ever test-ing for HIV.

**Data Availability Statement:** The data for the 2013 Philippine National Demographic and Health Survey Individual Recode are available from the Demographic and Health Surveys Program

Website (https://www.dhsprogram.com/data/available-datasets.cfm)

**Funding:** The authors have not received specific funding to conduct the analysis; however, they have received financial support from the Ateneo de Manila University School of Medicine and Public Health and the PLOS Publication Fee Assistance Office for the publication fee of the manuscript. These funding agencies did not have a role in the analysis, writing of the manuscript, as well as decision to publish.

**Competing interests:** The authors have declared that no competing interests exist.

## Conclusions

The low percentage of respondents who test for HIV is a call to further strengthen efforts to promote HIV testing among Filipino women. Information on its determinants can be used to guide the crafting and implementation of interventions to promote HIV testing to meet DOH and UNAIDS targets.

## Introduction

Despite the worldwide decrease in the incidence of Human Immunodeficiency Virus (HIV) infections [1,2], the Philippines is currently experiencing a rapid increase in the number of HIV cases [2–5]. For the first seven months of 2019, around 35 new cases of HIV are diagnosed in the country every day. From 1984 to July 2019, there have been 69,512 HIV cases that have been diagnosed in the Philippines; 4,339 (6.7%) of whom are women [6]. However, HIV statistics in the Philippines are perceived to be underestimates due to Filipinos' low knowledge and/or stigma associated with HIV testing [3–5,7,8]. It is estimated that around one-third of all Filipinos who have HIV do not know their true HIV status, despite HIV testing being free in many facilities throughout the country [3]. From the 2013 Philippine National Demographic and Health Survey (NDHS), only 2.3% of all the female respondents have reported that they have ever tested for HIV [9].

HIV testing is considered to be among the cornerstones of most HIV prevention and control strategies [10–12]. At the individual level, HIV testing, together with counselling, is an avenue where people can be educated about risky behaviors associated with the disease [13]. For those who have the disease, HIV testing is the first step into the continuum of care where they can be managed accordingly which will hopefully stop disease progression and transmission [12,14]. From a public health perspective, the greater the number of individuals who will undergo HIV testing, the more accurate the statistics will be for the disease. This will lead to better allocation of resources for public health interventions that will help curb the HIV epidemic [3,12]. For women, HIV testing has an added benefit of possibly preventing mother-to-child transmission of HIV. It is for this reason, together with the increasing numbers of pregnant women diagnosed with HIV and children born with HIV from 2011–16, that the Philippine Department of Health (DOH) has strongly encouraged pregnant women in the Philippines to undergo HIV testing. In relation to this, the DOH has decreed that by 2022, the proportion of people living with HIV (PLWH) who knows their status should be 90% [3]. This is in-line with the Joint United Nations Programme on HIV/AIDS (UNAIDS) 90-90-90 target, which stipulates that by 2020, "90% of all PLWH will know their true status, 90% of all people with diagnosed HIV infection will receive sustained antiretroviral therapy, and 90% of all people receiving antiretroviral therapy will have viral suppression" [15].

Given the importance of HIV testing among women, studies identifying its determinants have been carried out before. These determinants can be classified into socio-demographic determinants (e.g., age, educational attainment, address, religion, marital status, socio-economic status, employment, media exposure, and number of children) or HIV-related determinants (e.g., sexual behaviors, knowledge on HIV, perceptions on HIV testing, consumption of intoxicants, and having talked to mother or female guardian about HIV) [16–21]. Other determinants of HIV testing include having a dysfunctional relationship with their spouse/partner, tolerance of domestic violence, experiencing stigma, media exposure, number of lifetime sexual partners, having talked to mother/female guardian regarding HIV testing, ever pregnant,

and exposure to public health interventions regarding HIV [16,17,22]. Two reviews empha-sized that there are a host of social, institutional- and policy-level factors, often not considered in most observational studies, which may also act as barriers or enablers of HIV testing [23,24]. However, despite the numerous studies cited on HIV testing among women world-wide, and despite the HIV epidemic in the Philippines, there were no studies focusing on HIV testing among Filipino women in published literature. This is ostensibly due to the low propor-tion of cases of women with HIV in the country [6]. This implies that women could have been left behind in the response to the HIV epidemic in the country.

In order to address this gap and in order to craft interventions to encourage Filipino women to undergo testing, this analysis aims to identify the determinants of HIV testing among Filipino women. The results of this study could serve as the first step in the implemen-tation of interventions to promote HIV testing among Filipino women to help meet DOH and UNAIDS targets.

## Methods

### Study design, setting, and participants

This study is a secondary analysis of the 2013 Philippine NDHS women's individual recode data. The survey used a stratified two-stage sampling design with the 2010 Philippine Census of Population and Housing as sampling frame. The first stage sampling involved a systematic selection of 800 sample enumeration areas all over the country, distributed by urban/rural regions, to ensure representativeness. In the second stage, 20 housing units were randomly selected from each enumeration area using systematic sampling. All households in the sampled units were interviewed. From each household, women aged 15–49 were interviewed. The interviews were carried out all throughout the Philippines from August to October 2013. Other details of the sampling method for the 2013 Philippine NDHS can be found in its report [9].

### Data collection and study variables

The 2013 Philippine NDHS utilized a paper-based, pre-tested interview schedule to collect data on a wide range of socio-demographic, economic, knowledge on some health issues, health practices, fertility and childbirth, immunization of children, health insurance, domestic violence, women's empowerment, and other variables from a nationally-representative sample. A copy of the interview schedule can be seen on the final report of the 2013 Philippine NDHS [9].

Despite the multitude of variables collected in the study, only variables that are deemed to influence HIV testing were included in the analysis. The exposure variables for this study were: Age; educational attainment; civil status; condom use; consistent condom use; condom access; use of any traditional contraception method; tobacco consumption; age of husband/partner; educational attainment of partner; HIV knowledge, wealth index; address; tolerance to domestic-based gender violence; women's empowerment score; number of children; reli-gion, reading newspapers; weekly access to television, radio, newspapers, and internet; age of first sexual intercourse, and knowledge of condom source. The outcome variable for this study is HIV testing. A description of how the variables were operationally defined, as well as how they were coded are described in an Appendix (S1 Appendix).

To minimize observer bias, data collectors for the 2013 Philippine NDHS underwent a two-week training in administering the data collection tool. Furthermore, systematic random sam-pling was used to ensure representativeness. Moreover, data collectors visited the respondents at home repeatedly to ensure that the randomly selected respondents were interviewed, instead

of replacing them with whoever is convenient, thus minimizing selection bias. To minimize encoding errors, encoders underwent training in using the data entry program created specifically for this NDHS [9].

## Data management

Once permission was obtained from the NDHS data curators, the Individual Recode dataset of the 2013 Philippine NDHS was downloaded from the DHS website [25]. After this, the dataset was cleaned. In cleaning the dataset, new variables were generated from each variable that were included in the analysis. These new variables were cleaned and analysed to preserve the original data as much as possible. Inconsistent responses were considered as "no data" as the original responses of the respondents could no longer be obtained.

Some variables (e.g., employment status, marital status, etc.) were recoded to ensure that there were sufficient observations for each strata. Other variables (e.g., tobacco consumption) were recoded to ensure that the baseline stratum would have more observations, thus ensuring more stable estimates than if the current coding was used. Quantitative age variables were transformed into age brackets [e.g., 15–19, 20–24 years old, etc.] so that the effect of having similar ages on the outcome could be studied. The midpoint was assigned as the 'score' for each age group [e.g., the score '17' were assigned to those who were aged 15–19; the score '22' were assigned to those who were aged 20–24, etc.]. Condom use variables were recoded such that the baseline would be those who have never had sexual intercourse. Those who have used condoms consistently would also be noted with this variable. Similarly, variables on employment status or educational attainment of partner were recoded such that the baseline would be those who do not have partners at present.

Score variables (e.g., HIV knowledge score, women's empowerment, tolerance to domestic violence) were aggregated from many questions. HIV knowledge score were derived from the following questions: [1] Ever heard of AIDS; [2] Reduce risk of getting HIV: Always use condoms during sex; [3] Reduce risk of getting HIV: have one sex partner only, who has no other partners; [4] Can get HIV from mosquito bites; [5] Can get HIV by sharing food with person who has AIDS; [6] A healthy looking person can have HIV; and [7] Can get AIDS by shaking hands. Tolerance to domestic violence score was aggregated from the following questions: [1] Beating justified if wife goes out without telling husband; [2] Beating justified if wife neglects the children; [3] Beating justified if wife argues with husband; [4] Beating justified if wife refuses to have sex with husband; [5] Beating justified if wife burns the food. Women's empowerment score was derived from the following questions: [1] Who decides on your healthcare; [2] Who decides on large household purchases; [3] Who decides on daily household purchases; [4] Who decides on visits to family or relatives; and [5] Who decides what to do with money husband earns. For the HIV knowledge score questions, one point will be given for each correct answer, while no points will be given for incorrect or 'don't know' answers. For tolerance to domestic violence questions, one point will be given for each 'no' answer while no points will be given for 'don't know' answers. For each women empowerment questions, two points were given for each 'respondent only' answer, one point were given for each 'respondent and partner' answer and no points were given for each 'other answers'. The points from each question were added to come up with the HIV knowledge score, women's empowerment score, and tolerance to domestic violence score. A respondent with missing data in any of the questions that make up a score will not have an aggregate score. The aggregated score was left as a continuous variable so that the effect of a one-point increase in these variables on HIV testing can be quantified.

All data management and analyses were carried out in Stata/IC 14.0 [26].

## Data analysis

After preliminary cleaning, the dataset was declared as survey data and the sampling weights and strata (i.e., urban and rural, regions) were defined. All subsequent analyses, if applicable, were weighted. The distributions of each variable were determined by noting the respective histograms and measures of central tendency for continuous variables, and frequencies and proportions for categorical variables. For the descriptive analyses, weighted means and proportions will be shown; however, counts, medians, and modes will not be weighted.

The association of the exposures with HIV testing were examined using Pearson's $\chi^2$ test (for categorical exposure variables), adjusted Wald test (for normally-distributed continuous exposure variables), or the Wilcoxon rank-sum test (for skewed continuous exposure variables). The Pearson's $\chi^2$ test and the adjusted Wald test will be weighted; however, the Wilcoxon rank-sum test is not weighted because of the lack of applicable non-parametric statistical tests for weighted data. Those with missing data were not included in computing for the p-values for these tests. Crude odds ratios (OR) for each of the associations between exposure and the outcome were estimated using logistic regression for survey data.

Once the crude OR for this association were obtained, variables that might be in the causal pathway of other variables were excluded from the analyses. The remaining variables were then classified into whether they are proximal or distal risk factors. Proximal risk factors (PRFs) can be defined as factors that are thought to be closer to the outcome in a causal diagram, while distal risk factors (DRFs) were factors that were farther from the outcome and may indirectly contribute to causing it [27]. After this, a variable was generated to indicate respondents who do not have missing data for any of the remaining variables. Multivariate analyses were only carried out for respondents who have complete data for all of the variables of interest. To determine the order in which variables will be introduced into the final model, logistic regression for survey data was used to assess the effect of each PRF, adjusting for the DRFs with a p≤0.20 in the bivariate analyses. Adjusted OR of each PRF, as well as corresponding p-values were noted.

Logistic regression for survey data was used in the analyses of these associations. In building the final model for the determinants of HIV testing, DRFs were added into the model with the variable having the smallest p-value added first, then the second smallest p-value added second, and so on, until all DRFs with p≤0.20 from the bivariate analysis are in the model. After this, PRFs were added to the model starting with those with the smallest p-values in the analysis adjusting for DRFs until all the PRFs with p≤0.20 in the analyses adjusting for DRFs were added, or the maximum number of parameters was reached. While p-value cutoffs are not to be blindly followed in studying causal relationships in epidemiology, they may aid in variable selection to prevent models from being too overly-parameterized [28,29]. The maximum number of parameters for the final model are contingent on the effective sample size for the multivariate analysis, taking into consideration the 'rule of 10' events per parameter estimated [30].

At any point in the building of the final model, test for departure from the linearity assumption was carried out by observing the stratum-specific ORs, and running the *contrast* command in Stata once a quantitative ordinal variable (e.g., age group, wealth index, etc.) was added to the model. Since the midpoint of each age group was used as the 'score', parameters of a common linear trend would not only estimate the common linear effect of the age groups on the outcome, but also the common change in effect on the outcome per unit change in age [31]. In addition, model estimates were also observed for signs of multicollinearity or separation every time a variable is added. Variables with problematic estimates may be excluded from the analysis.

Considering that assessing effect measure modification (EMM) was not among the objectives, and that Mantel-Haenszel methods cannot be used in the analysis of survey data [32], no

assessment of EMM for any of the variables was carried out. Furthermore, no observations were deleted from the analyses to ensure that standard errors can be computed correctly [33]. Missing data were handled by presenting them in the univariate analyses and excluding respondents who have missing data in any of the variables of interest in the multivariate analyses.

Despite making several hypothesis tests, the level of significance was not adjusted. Instead, it was maintained at 0.05 all throughout the analysis as it is safer not to make adjustments for multiple comparisons in the analysis of empirical data to minimize errors in interpretation [34].

### Ethics

The 2013 Philippine NDHS has received ethical approval from ICF Macro Institutional Review Board (Project No.: 31561.00.000.00) dated July 1, 2010. This analysis has received ethical approval from the London School of Hygiene and Tropical Medicine MSc Ethics Committee (Reference No.: 15014).

### Results

The 2013 Philippine NDHS collected data from 16,437 Filipino women aged 15–49 years old. Interviews were completed for 16,155 individuals, with a 98.3% response rate. Except for counts, ranges, and non-parametric results, subsequent statistics shown are all weighted.

Only 372 (2.4%) respondents have ever tested for HIV. Most of the respondents finished secondary education, are married, do not use condom, do not use traditional contraception, are Roman Catholic, and have weekly television access. However, a substantial proportion of respondents have no data on condom access, age group of partner, and educational attainment of partner. This is predominantly because they have not had any sexual partners yet and/or have not had a partner at present. Among the categorical exposure variables and without adjusting for confounding, age of respondent, educational attainment of respondent, employment status of respondent, civil status, age at first intercourse, condom use, condom access, knowledge of condom source, usage of traditional contraception, tobacco use, educational attainment of partner, socio-economic status, and newspaper, television, and internet access were found to be associated with having ever tested for HIV (Table 1). All of these factors are positively associated with having ever tested for HIV, except for condom access and condom source. The negative association of these latter two variables with HIV testing denote that not having condom access and not knowing a condom source is a determinant of never testing for HIV.

Around 38% of the respondents have never had sexual intercourse, and majority do not have more than one sexual partner throughout their lifetime. Imputed age at first intercourse ranged from 7 to 47 years old. There are 5,891 (37.0) respondents who do not have children, and around 4,480 (28.3%) having only one or two children. Most of the respondents have a high (≥5/7) HIV knowledge score, have a high women empowerment score (≥6/10), and a low tolerance to domestic violence. The distributions of the number of lifetime sexual partners and HIV knowledge score were found to differ between those who were tested for HIV and those who were never tested for HIV. Despite these, none of the quantitative exposure variables had shown a strong evidence of association with HIV testing (Table 2).

For the multivariate analysis, distal risk factors that have a p≤0.20 in the cross-tabulations are age of respondent, highest educational attainment of respondent, employment status, civil status, tobacco use, highest educational attainment of partner, socio-economic status, domicile, religion, newspaper access, television access, and internet access. Proximal risk factors

**Table 1. Description of study participants and crude associations between categorical exposure variables and HIV testing (n = 16,155).**

| Variable | Never tested for HIV | Ever tested for HIV | χ² p-value | OR and 95% CI | p-value |
|---|---|---|---|---|---|
| Age group of respondent | | | <0.01 | | |
| 15–19 | 3,249 (99.6) | 12 (0.4) | | 1 | |
| 20–24 | 2,749 (97.9) | 60 (2.1) | | 5.96 (3.03–11.72) | <0.01 |
| 25–29 | 2,107 (96.8) | 64 (3.2) | | 9.36 (4.99–17.55) | <0.01 |
| 30–34 | 2,135 (96.7) | 71 (3.3) | | 9.62 (5.28–17.52) | <0.01 |
| 35–39 | 1,907 (96.5) | 67 (3.5) | | 10.24 (5.50–19.06) | <0.01 |
| 40–45 | 1,869 (97.6) | 47 (2.4) | | 6.86 (3.52–13.37) | <0.01 |
| 45–49 | 1,767 (97.1) | 51 (2.9) | | 8.29 (4.42–15.55) | <0.01 |
| Highest educational attainment of respondent | | | <0.01 | | |
| No education or primary education | 3,041 (99.0) | 26 (1.0) | | 1 | |
| Secondary education | 7,637 (98.5) | 110 (1.5) | | 1.46 (0.94–2.28) | 0.09 |
| Tertiary education or higher | 5,105 (95.6) | 236 (4.4) | | 4.51 (3.01–6.75) | <0.01 |
| Employment status of respondent | | | <0.01 | | |
| Unemployed | 8,265 (98.1) | 150 (1.9) | | 1 | |
| Currently employed | 7,516 (97.1) | 222 (2.9) | | 1.59 (1.28–1.97) | <0.01 |
| No data | 2 (100.0) | 0 (0.0) | | | |
| Civil status | | | <0.01 | | |
| Never in union | 5,427 (98.4) | 85 (1.6) | | 1 | |
| Married | 7,463 (97.6) | 182 (2.4) | | 1.54 (1.16–2.06) | <0.01 |
| Living with partner | 2,152 (96.8) | 69 (3.2) | | 2.05 (1.43–2.93) | <0.01 |
| Widowed/Divorced/Separated | 741 (95.3) | 36 (4.7) | | 3.07 (2.06–4.56) | <0.01 |
| Age at first intercourse | | | <0.01 | | |
| Never had any sexual partner | 6,043 (98.3) | 104 (1.7) | | 1 | |
| ≤19 | 4,810 (97.6) | 113 (2.4) | | 1.42 (1.05–1.91) | 0.02 |
| 20–24 | 3,325 (97.0) | 98 (3.0) | | 1.74 (1.28–2.36) | <0.01 |
| 25–29 | 1,132 (96.6) | 42 (3.4) | | 2.02 (1.34–3.03) | <0.01 |
| 30+ | 353 (96.6) | 14 (3.4) | | 2.02 (1.09–3.72) | 0.03 |
| No data | 120 (99.4) | 1 (0.6) | | | |
| Condom use | | | <0.01 | | |
| Never had any sexual partner | 6,043 (98.3) | 104 (1.7) | | 1 | |
| Did not use condom with last sexual partner | 9,516 (97.3) | 260 (2.7) | | 1.59 (1.23–2.06) | <0.01 |
| Used condom with last sexual partner but uses inconsistently | 37 (97.1) | 1 (2.9) | | 1.68 (0.23–12.55) | 0.61 |
| Consistent condom use with last sexual partner | 171 (95.4) | 7 (4.6) | | 2.74 (1.28–5.86) | <0.01 |
| No data | 16 (100.0) | 0 (0.0) | | | |
| Condom access | | | <0.01 | | |
| Respondent can get a condom | 8,135 (96.7) | 270 (3.3) | | 1 | |
| Respondent cannot get a condom | 4,131 (98.1) | 80 (1.9) | | 0.56 (0.43–0.73) | <0.01 |
| No data | 3,517 (99.4) | 22 (0.6) | | | |
| Knowledge of condom source | | | <0.01 | | |
| Knows any source of condom | 12,363 (97.2) | 355 (2.8) | | 1 | |
| Does not know any source of condom | 3,418 (99.5) | 17 (0.6) | | 0.19 (0.11–0.32) | <0.01 |
| No data | 2 (100.0) | 0 (0.0) | | | |
| Traditional or folkloric contraception | | | 0.03 | | |
| Does not use traditional or folkloric contraception | 14,115 (97.7) | 321 (2.3) | | 1 | |
| Uses traditional or folkloric contraception | 1,668 (96.8) | 51 (3.2) | | 1.43 (1.03–1.98) | 0.03 |

(*Continued*)

**Table 1.** (Continued)

| Variable | Never tested for HIV | Ever tested for HIV | χ² p-value | OR and 95% CI | p-value |
|---|---|---|---|---|---|
| Tobacco use | | | <0.01 | | |
| Non-user | 14,881 (97.8) | 319 (2.2) | | 1 | |
| User | 902 (94.4) | 53 (5.6) | | 2.69 (1.90–3.82) | <0.01 |
| Age group of partner | | | 0.29 | | |
| 15–24 | 807 (97.7) | 19 (2.3) | | 1 | |
| 25–29 | 1,340 (98.1) | 26 (1.9) | | 0.82 (0.45–1.50) | 0.52 |
| 30–34 | 1,681 (96.6) | 52 (3.4) | | 1.47 (0.87–2.50) | 0.15 |
| 35–39 | 1,722 (97.6) | 44 (2.4) | | 1.04 (0.59–1.83) | 0.89 |
| 40–45 | 1,670 (97.1) | 49 (2.9) | | 1.26 (0.72–2.20) | 0.42 |
| 45–49 | 1,309 (97.6) | 32 (2.4) | | 1.04 (0.56–1.92) | 0.90 |
| 50+ | 1,086 (97.3) | 29 (2.7) | | 1.16 (0.63–2.15) | 0.63 |
| *No data* | 6,168 (98.0) | 121 (2.0) | | | |
| Highest educational attainment of partner | | | <0.01 | | |
| No education or primary education | 3,179 (98.6) | 39 (1.4) | | 1 | |
| Secondary education | 4,218 (97.6) | 103 (2.4) | | 1.73 (1.16–2.58) | <0.01 |
| Tertiary education or higher | 2,937 (95.5) | 143 (4.5) | | 3.32 (2.26–4.87) | <0.01 |
| *No data* | 5,449 (98.4) | 87 (1.6) | | | |
| Wealth index | | | <0.01 | | |
| Poorest | 3,177 (99.4) | 17 (0.6) | | 1 | |
| Poorer | 3,050 (98.9) | 37 (1.2) | | 1.88 (1.00–3.51) | 0.05 |
| Middle | 3,060 (97.8) | 68 (2.2) | | 3.57 (2.10–6.09) | <0.01 |
| Richer | 3,185 (97.2) | 101 (2.8) | | 4.62 (2.75–7.77) | <0.01 |
| Richest | 3,311 (95.7) | 150 (4.3) | | 7.19 (4.37–11.82) | <0.01 |
| Domicile | | | <0.01 | | |
| Urban | 7,412 (97.4) | 197 (2.6) | | 1 | |
| Rural | 8,371 (97.9) | 175 (2.1) | | 0.79 (0.61–1.02) | 0.07 |
| Religion | | | 0.10 | | |
| Roman Catholicism | 11,799 (97.7) | 279 (2.3) | | 1 | |
| Other Christian denomination | 1,444 (97.4) | 32 (2.6) | | 1.12 (0.76–1.65) | 0.56 |
| Islam | 1,331 (98.5) | 15 (1.5) | | 0.65 (0.39–1.10) | 0.11 |
| None/other beliefs | 1,193 (96.7) | 39 (3.3) | | 1.42 (0.96–2.10) | 0.07 |
| *No data* | 16 (100.0) | 0 (0.0) | | | |
| Newspaper access | | | <0.01 | | |
| None or less than once a week | 11,759 (97.9) | 237 (2.1) | | 1 | |
| More than once a week | 4,016 (96.8) | 135 (3.2) | | 1.33 (1.17–1.53) | <0.01 |
| *No data* | 8 (100.0) | 0 (0.0) | | | |
| Television access | | | <0.01 | | |
| None or less than once a week | 3,520 (99.0) | 33 (1.0) | | 1 | |
| More than once a week | 12,242 (97.3) | 339 (2.7) | | 2.72 (1.86–3.97) | <0.01 |
| *No data* | 21 (100.0) | 0 (0.0) | | | |
| Radio access | | | 0.16 | | |
| None or less than once a week | 7,636 (97.8) | 160 (2.2) | | 1 | |
| More than once a week | 8,117 (97.5) | 212 (2.6) | | 1.17 (0.94–1.46) | 0.16 |
| *No data* | 30 (100.0) | 0 (0.0) | | | |
| Internet access | | | <0.01 | | |
| None or less than once a week | 11,459 (98.3) | 186 (1.7) | | 1 | |
| More than once a week | 4,258 (96.0) | 185 (4.0) | | 2.48 (2.00–3.08) | <0.01 |
| *No data* | 66 (97.9) | 1 (2.1) | | | |

**Table 2. Description of study participants and crude associations between quantitative exposures and HIV testing.**

| Variable | Number of respondents with data | Range | Mean and 95% Confidence Interval | Median | Distribution | Rank-sum test p-value | OR[a] | p-value |
|---|---|---|---|---|---|---|---|---|
| Number of children | 16,155 (100) | 0–19 | 2.06 (2.01–2.11) | 1 | Right-skewed | 0.07 | 1.00 (0.96–1.04) | 0.91 |
| Number of lifetime sexual partners | 16,145 (99.9) | 0–95 | 0.76 (0.74) | 1 | Right-skewed | <0.01 | 1.14 (0.95–1.37) | 0.15 |
| HIV knowledge score | 14,607 (90.4) | 1–7 | 4.53 (4.51–4.57) | 5 | Left-skewed | 0.02 | 1.08 (0.98–1.18) | 0.10 |
| Tolerance to domestic violence score | 16,144 (99.9) | 0–5 | 0.26 (0.24–0.28) | 0 | Right-skewed | 0.52 | 0.98 (0.88–1.11) | 0.80 |
| Women's empowerment score | 9,456 (58.5) | 0–10 | 6.50 | 6 | Left-skewed | 0.68 | 1.03 (0.95–1.12) | 0.52 |

[a]Denote increase in odds of HIV testing per unit increase in the value of the quantitative exposure variable.

that have a p≤0.20 in the cross-tabulations are age at first intercourse, condom use, condom access, knowledge of condom source, traditional contraception, number of children, number of lifetime sexual partners and HIV knowledge score. However, because there is collinearity between knowledge of condom source and condom access, and because the latter has a lot of missing data, it will not be among the variables that will be considered in the analysis. Only 8,578 (53.2%) respondents have complete data for the variables that are considered in the multivariate analysis. Out of these, 243 (2.8%) have underwent HIV testing (Table 3).

In building the final model, tests for linear trend were run for age of respondent, age at first sexual intercourse, and socio-economic status. Age of respondent (p = 0.27) and age at first sexual intercourse (p = 0.92) did not show evidence of deviation from a linear trend, but there is an evidence for deviation of a linear trend for socio-economic status (p<0.01), which meant that stratum-specific ORs were shown for socio-economic status instead of common ORs.

After adjusting for other variables, having tertiary education (adjusted odds ratio (aOR) = 2.15; 95% Confidence Interval (CI): 1.15–4.04), being unmarried but living together with partner (aOR = 1.72; 95% CI: 1.19–2.48), tobacco use (aOR = 1.87; 95% CI: 1.13–3.11); belonging to the middle class (aOR = 2.72; 95% CI: 1.30–5.67), richer (aOR = 3.00; 95% CI: 1.37–5.68), and richest (aOR = 4.14; 95% CI: 1.80–5.91) populations, having weekly television access (aOR = 1.75; 95% CI: 1.04–2.94) or internet access (aOR = 2.01; 95% CI: 1.35–3.00), living in a rural area (aOR = 1.87; 95% CI: 1.34–2.61); and being a Muslim (aOR = 2.30; 95% CI: 1.15–4.57) were associated with higher odds of HIV testing among Filipino women aged 15–49.

## Discussion

Only around 2% of Filipino women have had HIV testing throughout their lifetimes, implying that there is still substantial work to be done in promoting HIV testing to Filipino women to meet DOH and UNAIDS targets. Women's educational attainment, civil status, tobacco use, socio-economic status, television and internet access, domicile, and religion showed strong evidence of association with HIV testing. This information could be used to guide the development of interventions to promote HIV testing among Filipino women.

These associations were similar to the findings of other studies. Specifically, there seems to be an increasing propensity for HIV testing among more educated or wealthier respondents, regardless of gender [7,16]. A study conducted in the United States also found that smoking was found to be strongly associated with HIV testing. Accordingly, the said study explains that smokers might be more likely to undergo HIV testing because being a smoker is associated

**Table 3.  Determinants of HIV testing among Filipino women (n = 8,578).**

| | Adjusted[a] OR and 95% CI | p-value |
|---|---|---|
| Age of respondent | 1.02 (1.00–1.05)[b] | 0.09 |
| Educational attainment | | |
| No education or primary education | 1 | |
| Secondary education | 1.26 (0.67–2.38) | 0.48 |
| Tertiary education or higher | 2.15 (1.15–4.04) | 0.02 |
| Employment status | | |
| Unemployed | 1 | |
| Currently employed | 0.99 (0.73–1.34) | 0.95 |
| Civil status | | |
| Married | 1 | |
| Living with partner | 1.72 (1.19–2.48) | <0.01 |
| Widowed/Divorced/Separated | 1.48 (0.60–3.67) | 0.40 |
| Tobacco use | | |
| Non-user | 1 | |
| User | 1.87 (1.13–3.11) | 0.02 |
| Educational attainment of partner | | |
| No education or primary education | 1 | |
| Secondary education | 0.88 (0.54–1.45) | 0.62 |
| Tertiary education or higher | 0.84 (0.50–1.44) | 0.53 |
| Socio-economic status | | |
| Poorest | 1 | |
| Poorer | 1.48 (0.68–3.21) | 0.32 |
| Middle | 2.72 (1.30–5.67) | <0.01 |
| Richer | 3.00 (1.37–6.58) | <0.01 |
| Richest | 4.14 (1.80–9.51) | <0.01 |
| Newspaper access | | |
| None or less than once a week | 1 | |
| More than once a week | 0.85 (0.60–1.19) | 0.34 |
| Television access | | |
| None or less than once a week | 1 | |
| More than once a week | 1.75 (1.04–2.94) | 0.04 |
| Internet access | | |
| None or less than once a week | 1 | |
| More than once a week | 2.01 (1.35–3.00) | <0.01 |
| Domicile | | |
| Urban | 1 | |
| Rural | 1.87 (1.34–2.61) | <0.01 |
| Religion | | |
| Roman Catholicism | 1 | |
| Other Christian denomination | 1.08 (0.66–1.77) | 0.77 |
| Islam | 2.30 (1.15–4.57) | 0.02 |
| None/other beliefs | 1.17 (0.68–2.04) | 0.57 |
| Age at first sexual intercourse | 0.99 (0.97–1.02)[b] | 0.59 |
| Condom use | | |
| Did not use condom with last sexual partner | 1 | |
| Used condom with last sexual partner but uses inconsistently | 1.13 (0.13–9.71) | 0.91 |
| Consistent condom use with last sexual partner | 0.80 (0.30–2.19) | 0.67 |

(*Continued*)

**Table 3.** (Continued)

| | Adjusted[a] OR and 95% CI | p-value |
|---|---|---|
| Knowledge of condom source | | |
| Knows any source of condom | 1 | |
| Does not know any source of condom | 0.64 (0.34–1.21) | 0.17 |
| Traditional or folkloric contraception | | |
| Does not use traditional or folkloric contraception | 1 | |
| Uses traditional or folkloric contraception | 1.22 (0.85–1.75) | 0.29 |
| HIV Knowledge | 0.96 (0.85–1.10)[b] | 0.56 |
| Number of children | 0.99 (0.90–1.09)[b] | 0.85 |
| Number of lifetime sexual partners | 1.08 (0.97–1.20)[b] | 0.18 |

[a]Adjusted for other variables listed in this table.

[b]Denote increase in odds of HIV testing per unit increase in the value of the quantitative exposure variable.

with risky sexual behaviors and/or drug use, the latter two are known independent risk factors for HIV [35]. Due to certain religious taboos, HIV testing remains very low among some religious groups in the country. However, the odds of HIV testing are highest among Muslims. While there are no studies explaining this phenomenon in the Philippines, a study conducted in Malaysia explains that in their country, Muslim religious leaders were supportive of HIV testing because it provides a protective mechanism in line with Islamic teachings [36]. The specifics of the association between media exposure and HIV testing was examined in detail in this study and was found to be similar to those that are found in other settings [16,17]. Frequent exposure to television and Internet also increases the probability of exposure to HIV information, education, and communication (IEC) campaigns promoting HIV testing disseminated through these forms of media, thus promoting HIV testing.

There were also differences in the findings of this study with what has been published in literature. In this analysis, older individuals were found to be more likely to have undergone HIV testing than younger respondents, but this trend is the exact opposite of what was found in Burkina Faso, where older women were found to be less likely to test than younger ones. The same study in Burkina Faso found that living in a rural area inhibits HIV testing [16], while this analysis found that those from rural areas are more likely to have undergone HIV testing as compared to those from urban areas. Without adjusting for confounders, we found several factors to be associated with HIV testing in this analysis, but a secondary analysis of data collected on 2003 from Filipino males show that only HIV knowledge is strongly associated with getting HIV test result [7].

While consistency of results across populations or circumstances strengthen evidence for causation [37], its absence does not necessarily mean that results are no longer valid nor useful. A possible reason explaining the differences in the effect of age on HIV testing is the difference in how age was handled in the analyses. This study grouped respondents on five-year age groups, while other studies grouped respondents on 10-year groups [16,22]. Another possible reason for the differences between the findings of this study and others is that the populations and contexts on the studies being compared might be inherently different. Differences in social, economic and political context underpinning HIV epidemiology and response should not be ignored in comparing findings from different settings [38–41]. Findings from the older study involving Filipino males may differ from the current study due to gender differences. Secular changes may also explain why results differed between the previous study and this analysis [7].

The study presents several salient points of concern. First, the prevalence of HIV testing remains to be very low. Second, the association of socio-economic status and highest educational attainment with HIV testing highlights inequities in access and utilization of HIV testing services, despite it being offered for free in government facilities. This is ostensibly explained by low awareness of HIV testing, and an even lower awareness that it is offered for free [3]. Third, the Philippine DOH has made significant strides to encourage HIV testing among pregnant women [3], but as the results show, number of children was not found to be associated with HIV testing which highlight the need to do more in promoting HIV testing among pregnant women. Fourth, the lower odds of testing among those who are from urban areas are worrying because urban centers in the Philippines are where HIV cases are rapidly rising.

Despite these worrying conclusions, the study is best interpreted with its limitations in mind. The exclusion of almost half of the respondents in the multivariate analysis due to missing data underlines the possibility of selection bias. The respondents who were excluded were mostly those who do not have partners, or have never had sexual intercourse, because these respondents did not have data for educational attainment of partner. The exclusion of these respondents also meant that the baseline for the condom use variable are no longer those that have never had intercourse, as in the univariate analysis, but those who did not use condom in their last intercourse. This also meant that the baseline for the civil status variable are now those who are married, instead of those who were never in union as in the univariate analysis. A separate model was considered for those who do not have partners or those who never had sexual intercourse, but the very low proportion of respondents who tested for HIV for these populations meant that such a model might have low statistical power. Not to mention, those who never had sexual intercourse is deemed to have low risk in developing HIV as HIV is mostly transmitted sexually here in the Philippines. Given this, it should be kept in mind that the findings of this analysis may only be generalized to those who have already had sexual partners.

Alternative variable selection strategies emphasize that all known confounders should be controlled for in the model [42]. From this line of reasoning, there would still be residual confounding as we have not controlled for variables either because they were not collected in the original dataset (i.e., social support, drug use, etc. and other factors working beyond the individual level), or were excluded due to the specified p-value cutoff in the Methodology. However, controlling for all known confounders might lead to overly parameterized models, especially that our proportion of HIV testers is very low. It is for this reason that p-value cutoffs were used to select variables to include in the model. Even the multivariate model itself fails to meet the 'rule-of-10', having estimated 29 parameters on 243 events (i.e., people who tested for HIV), giving us 8.4 events per parameter. However, simulation studies have shown that the 'rule-of-10' can be relaxed to up to five events per parameter without expecting issues in chances of type-I error, problematic confidence intervals, and high relative bias [30].

Cross-sectional studies such as this analysis are especially susceptible to reverse causality, especially for data that may vary with time. This is often a problem for this study design as both exposure and outcome data are collected simultaneously. This prevents ascertainment of the temporal direction of the associations found in the study [43].

Another issue that usually affect HIV studies using self-report data, including this analysis, is response bias [44]. This was apparent for age at first sexual intercourse, which necessitated the use of imputed data. This also implies that sexual behavior (e.g., condom use, etc.) and other health data collected from the respondents should be interpreted cautiously due to the possibility of Hawthorne effect [45]. Ultimately, this implies that conclusions drawn from this analysis is only as good as the quality of data provided by the respondents.

Most importantly, there have been developments in HIV testing in the Philippines since the data was collected on 2013. On 2016, the country has piloted rapid diagnostic screening tests among high-burden cities in the country to increase uptake of HIV testing. These rapid diagnostic tests have the advantage of being cheaper and having a faster turn-around time as compared to current Western blot-based confirmatory tests [3,46,47]. However, despite the rollout of these initiatives, HIV testing remains very low and falls short of the 90-90-90 target set by the DOH and UNAIDS [3]. On 2019, the country has started the implementation of the new Philippine HIV and AIDS Policy Act. Among the provisions of this new law is allowing persons aged 15–18 to undergo HIV testing without parental consent and allowing pregnant and other adolescents younger than 15 years old and engaging in high-risk behavior to undergo testing without parental consent [48]. Owing to its recent implementation, however, we are yet to measure how this new law affects uptake and utilization of HIV testing, especially among Filipino women.

Despite these weaknesses and the policy changes since the data was collected, these findings should still be considered in formulating public health interventions to promote HIV testing, considering the dearth of evidence exploring this phenomenon and the urgency of the HIV situation in the Philippines. Further research should be undertaken to elucidate the relationships of some exposures with HIV testing to improve on the weaknesses of this study as well as assess the effect of new policy developments on uptake and utilization of HIV testing among Filipino women.

## Conclusions

The low proportion of Filipino women who have ever tested for HIV is a call to strengthen efforts to promote HIV testing. Information on its determinants can help in the formulation and implementation of interventions and which segments of the population should be targeted by these interventions. Information, education, and communication campaigns to promote HIV testing and to dispel myths surrounding it should be disseminated via television or Internet. Such campaigns should target those who have lower socio-economic status, those who have low educational attainments, and those who live in urban areas. Further research to identify determinants of HIV testing, especially among populations that were not studied yet, should be done to identify segments of the population that should be reached by interventions to promote HIV testing. Further research to assess the impact of recent policies on HIV testing should likewise be conducted. Studies and implementation research focusing on availability, accessibility, and acceptability of HIV testing, including novel and alternative approaches, such as self-testing [46,49] and use of technology [50] should likewise be conducted. Only through the promotion of HIV testing, and its subsequent uptake by the population, will the DOH and UNAIDS reach their targets for the Philippines.

## Supporting information

**S1 Appendix. Definition of variables and coding manual.**
(DOCX)

## Acknowledgments

We thank the DHS Program for lending us the 2013 Philippine National Demographic and Health Survey dataset. We are also grateful for the comments of Ms. Arianna Maever L. Amit and anonymous reviewer/s from the London School of Hygiene and Tropical Medicine for improving this manuscript.

## Author Contributions

**Conceptualization:** Veincent Christian F. Pepito, Sam Newton.

**Formal analysis:** Veincent Christian F. Pepito.

**Methodology:** Veincent Christian F. Pepito, Sam Newton.

**Supervision:** Sam Newton.

**Writing – original draft:** Veincent Christian F. Pepito.

**Writing – review & editing:** Sam Newton.

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
