## [Decision Letter · Decision Letter 0]

24 Mar 2020

PONE-D-20-02945

Determinants of HIV Testing among Filipinas: Results from the 2013 Philippine National Demographic and Health Survey

PLOS ONE

Dear Mr Pepito,

Thank you for submitting your manuscript to PLOS ONE. After careful consideration, we feel that it has merit but does not fully meet PLOS ONE’s publication criteria as it currently stands. Therefore, we invite you to submit a revised version of the manuscript that addresses the points raised during the review process.

We would appreciate receiving your revised manuscript by May 08 2020 11:59PM. To enhance the reproducibility of your results, we recommend that if applicable you deposit your laboratory protocols in protocols.io, where a protocol can be assigned its own identifier (DOI) such that it can be cited independently in the future. For instructions see: http://journals.plos.org/plosone/s/submission-guidelines#loc-laboratory-protocols

We look forward to receiving your revised manuscript.

Kind regards,

Joel Msafiri Francis, MD, MS, PhD

Academic Editor

PLOS ONE

Journal Requirements:

Reviewers' comments:

Reviewer's Responses to Questions

**Comments to the Author**

1. Is the manuscript technically sound, and do the data support the conclusions?

Reviewer #1: Yes

Reviewer #2: Yes

2. Has the statistical analysis been performed appropriately and rigorously? 

Reviewer #1: Yes

Reviewer #2: Yes

3. Have the authors made all data underlying the findings in their manuscript fully available?

Reviewer #1: Yes

Reviewer #2: Yes

4. Is the manuscript presented in an intelligible fashion and written in standard English?

Reviewer #1: Yes

Reviewer #2: Yes

5. Review Comments to the Author

Reviewer #1: 1. Tables 1 and 3 are largly overlapped. Accordingly, merge them into one table.

2. Add the level of reference for OR in Table 4, like Table 5.

3. In Table 5, the reference of OR is not "1.00", but "1", because it is not an estimate.

4. In the footnote of Table 5, explain or list the factors used for adjustment of OR.

5. Correct upper/lower case of the first letter of words in the title of paper in References according to the rule of this journal. It is inconsistent.

Reviewer #2: The secondary analysis presented in this paper is of importance to the Philippines context given the rapid increase in HIV cases and the current focus on sex-workers, cis-men who have sex with men (MSM) and transgender women (TGW) which could leave female populations neglected in the HIV response. The age of the dataset used does raise some concerns given culture is dynamic and evolves with time and HIV services have undergone substantial reform in the Philippines. However, available evidence on factors associated with HIV testing among Filipinas is very limited and therefore analysis of this 2013 dataset is somewhat justified. The authors could do more to argue why this evidence gap should be filled.

I commend the authors for the detailed limitations presented in the discussion and recommendations on how findings from the analysis can be taken forward to inform further research. Specific comments for the author’s consideration are numbered below:

1. Readers unfamiliar with the Philippines may not understand the gendered term Filipinas so would advise providing a definition.

2. Lines 48-50: The first sentence references the reported low rate of women diagnosed with HIV. The second claims that this is likely to be an underestimate due to low knowledge and misconceptions towards HIV testing among Filipinos. However, the references included (4,5,7) are commentary pieces/ focused on Filipino men. There potentially isn’t published evidence available to assess if the number of Filipino women diagnosed is an underestimate. I therefore recommend removing the second sentence and move references (4 and 5) to the paragraph starting on row 70.

3. The authors may want to consider including the following reference given the findings presented in table 4:

Marguerite B. Lucea, Michelle J. Hindin, Joan Kub & Jacquelyn C. Campbell (2012) HIV Risk, Partner Violence, and Relationship Power Among Filipino Young Women: Testing a Structural Model, Health Care for Women International, 33:4, 302-320, DOI: 10.1080/07399332.2011.646369

4. It’s worth noting that the Philippines HIV testing strategies has undergone substantial reform since 2013. In 2016, the National HIV/AIDS and STI Prevention and Control Program introduced a rapid HIV diagnostic algorithm (rHIVda); expected to reduce turnaround time for results from 7-21 days to 30-60 minutes. Between 2016-2017 rHIVda was piloted and validated in 8 government and non-government VCT clinics in the National Capital Region and Davao Region. Since then efforts have been made to roll out rHIVda to facilities across the country. However, testing services are predominantly targeted towards cis-MSM, sex-workers and TGW which could lead female populations to believe free rapid diagnostics are not appropriate or accessible to them.

Given factors associated with the availability, accessibility and acceptability of HIV testing services has previously been found to affect HIV testing uptake in female populations this is important contextual information to include, and strengthens the argument for further research to address the limitations in this study.

6. PLOS authors have the option to publish the peer review history of their article (what does this mean?). If published, this will include your full peer review and any attached files.

Reviewer #1: Yes: Nobuyuki Hamajima

Reviewer #2: Yes: Charlotte Hemingway

---

## [Author Response · Author response to Decision Letter 0]

14 Apr 2020

Dr. Nobuyuki Hamajima's comments:

1. Tables 1 and 3 are largly overlapped. Accordingly, merge them into one table.

Response: We have merged Tables 1 and 3 into Table 1 (p. 11-16). Realizing that Tables 2 and 4 are similar, we have merged it as well into Table 2 (p. 17-18). 

2. Add the level of reference for OR in Table 4, like Table 5.

Response: The ORs in Table 4 (now in Table 2, p. 17-18) are common ORs denoting an increase in the odds per unit increase in the quantitative exposure. The level of reference is the lowest value in the range. A footnote has been added clarifying this. 

3. In Table 5, the reference of OR is not "1.00", but "1", because it is not an estimate.

Response: We revised the baseline values to 1. See Table 3 (p. 18-21).

4. In the footnote of Table 5, explain or list the factors used for adjustment of OR.

Response: A footnote was added in Table 3 (p. 18-21) explaining that the adjusted ORs were obtained by adjusting for the other variables listed in the table. 

5. Correct upper/lower case of the first letter of words in the title of paper in References according to the rule of this journal. It is inconsistent.

Response: We revised this. The upper/lower case formatting of references is fixed according to PLOS guidelines. See References (p. 27-33).

Dr. Charlotte Hemingway's comments

The secondary analysis presented in this paper is of importance to the Philippines context given the rapid increase in HIV cases and the current focus on sex-workers, cis-men who have sex with men (MSM) and transgender women (TGW) which could leave female populations neglected in the HIV response. The age of the dataset used does raise some concerns given culture is dynamic and evolves with time and HIV services have undergone substantial reform in the Philippines. However, available evidence on factors associated with HIV testing among Filipinas is very limited and therefore analysis of this 2013 dataset is somewhat justified. The authors could do more to argue why this evidence gap should be filled.

Response: We agree that the age of the dataset is quite an issue. On a more practical note, when we did the analysis this was the most recent NDHS to date and we were not given access to other datasets, thus, we had to make do with what we have. To address this, we have substantially reworked my Introduction (p. 3-4) to focus solely on women and the lack of data on this HIV testing among Filipino women. We had also introduced a new paragraph in the Discussion on the recent developments of HIV testing in the Philippines, including the discussion of rHIVda that you have mentioned, as well as the relaxed guidelines for consent for HIV testing among youth (p. 25). 

1. Readers unfamiliar with the Philippines may not understand the gendered term Filipinas so would advise providing a definition.

Response: Revised. We decided that probably it is better to use the term “Filipino women” instead of “Filipina” as everybody is more familiar with this. 

2. Lines 48-50: The first sentence references the reported low rate of women diagnosed with HIV. The second claims that this is likely to be an underestimate due to low knowledge and misconceptions towards HIV testing among Filipinos. However, the references included (4,5,7) are commentary pieces/ focused on Filipino men. There potentially isn’t published evidence available to assess if the number of Filipino women diagnosed is an underestimate. I therefore recommend removing the second sentence and move references (4 and 5) to the paragraph starting on row 70.

Response: The underestimation of HIV statistics has been mentioned in other references such as reference 3 and the reference you gave me (Lucea et al). Lucea, in their article’s background particularly notes that HIV statistics for women in the Philippines is also most likely an underestimate. Thus, we have clarified this statement (p. 3) and included new references (i.e., Lucea et al) to substantiate our claim. 

3. The authors may want to consider including the following reference given the findings presented in table 4:

Marguerite B. Lucea, Michelle J. Hindin, Joan Kub & Jacquelyn C. Campbell (2012) HIV Risk, Partner Violence, and Relationship Power Among Filipino Young Women: Testing a Structural Model, Health Care for Women International, 33:4, 302-320, DOI: 10.1080/07399332.2011.646369

Response: Thank you very much for this reference. While we were unable to tie it to Table 4 (now Table 2) as their paper barely mentioned HIV testing after their Introduction, we were able to use it to improve my Introduction, specifically as they mentioned that HIV statistics for Filipino women are most likely underestimates. 

4. It’s worth noting that the Philippines HIV testing strategies has undergone substantial reform since 2013. In 2016, the National HIV/AIDS and STI Prevention and Control Program introduced a rapid HIV diagnostic algorithm (rHIVda); expected to reduce turnaround time for results from 7-21 days to 30-60 minutes. Between 2016-2017 rHIVda was piloted and validated in 8 government and non-government VCT clinics in the National Capital Region and Davao Region. Since then efforts have been made to roll out rHIVda to facilities across the country. However, testing services are predominantly targeted towards cis-MSM, sex-workers and TGW which could lead female populations to believe free rapid diagnostics are not appropriate or accessible to them.

Response: Revised. We have introduced a paragraph in the Discussion mentioning new developments to HIV testing in the country; specifically, on rHIVda and relaxed guidelines for consent (p. 25). 

Given factors associated with the availability, accessibility and acceptability of HIV testing services has previously been found to affect HIV testing uptake in female populations this is important contextual information to include and strengthens the argument for further research to address the limitations in this study.

Response: Revised our Discussion and Conclusion for this. In our conclusion, we specifically highlighted the need for research on availability, accessibility and acceptability of HIV testing (p. 26).

---

## [Decision Letter · Decision Letter 1]

20 Apr 2020

Determinants of HIV Testing among Filipino women: Results from the 2013 Philippine National Demographic and Health Survey

PONE-D-20-02945R1

Dear Dr. Pepito,

We are pleased to inform you that your manuscript has been judged scientifically suitable for publication and will be formally accepted for publication once it complies with all outstanding technical requirements.

With kind regards,

Joel Msafiri Francis, MD, MS, PhD

Academic Editor

PLOS ONE

Additional Editor Comments (optional):

Reviewers' comments:

Reviewer's Responses to Questions

**Comments to the Author**

1. If the authors have adequately addressed your comments raised in a previous round of review and you feel that this manuscript is now acceptable for publication, you may indicate that here to bypass the “Comments to the Author” section, enter your conflict of interest statement in the “Confidential to Editor” section, and submit your "Accept" recommendation.

Reviewer #1: All comments have been addressed

Reviewer #2: All comments have been addressed

2. Is the manuscript technically sound, and do the data support the conclusions?

Reviewer #1: Yes

Reviewer #2: (No Response)

3. Has the statistical analysis been performed appropriately and rigorously? 

Reviewer #1: Yes

Reviewer #2: (No Response)

4. Have the authors made all data underlying the findings in their manuscript fully available?

Reviewer #1: Yes

Reviewer #2: (No Response)

5. Is the manuscript presented in an intelligible fashion and written in standard English?

Reviewer #1: Yes

Reviewer #2: (No Response)

6. Review Comments to the Author

Reviewer #1: The authors responded my comments and revised the manuscript adequately. There are no further comments.

Reviewer #2: (No Response)

7. PLOS authors have the option to publish the peer review history of their article (what does this mean?). If published, this will include your full peer review and any attached files.

Reviewer #1: Yes: Nobuyuki Hamajima

Reviewer #2: Yes: Charlotte Hemingway

---

## [Editor Report · Acceptance letter]

30 Apr 2020

PONE-D-20-02945R1 

Determinants of HIV Testing among Filipino women: Results from the 2013 Philippine National Demographic and Health Survey 

Dear Dr. Pepito:

I am pleased to inform you that your manuscript has been deemed suitable for publication in PLOS ONE. Congratulations! Your manuscript is now with our production department. 

With kind regards,

on behalf of

Dr. Joel Msafiri Francis 

Academic Editor

PLOS ONE